# ITEM LANGUAGE MODEL

## ABSTRACT

Embeddings are extensively used in many domains to represent information about domain entities in a compressed manner. In recommendation systems, these embeddings are trained to extract meaningful information about items or users from behavioral data consisting of users' ratings or users' implicit feedback. These behavioral embeddings are usually not trained on data from a language domain, but they encode useful behavioral information which cannot be easily described using language. In contrast, Large Language Models (LLMs) do not have good representations for either behavioral data or behavioral entities(items or users), as these are usually not textual and the data is specific to a recommendation system. Bridging this gap between behavioral understanding and language understanding can enable new item and language interleaved tasks. In our work we show how we can efficiently adapt rich behavioral embeddings for use as input representation in pre-trained LLMs. To achieve this we adapt a Querying Transformer with a new item-item contrastive loss and show improved item-text joint understanding in PALM-2 and also demonstrate improved capabilities in recommendation domain compared to using the behavioral embeddings directly as input to PALM-2.

## 1 INTRODUCTION

Large Language Models (LLMs), trained on web-scale data using a very large number of parameters, have shown remarkable emergent capabilities, such as in-context learning, reasoning, coding (Brown et al., 2020; Chowdhery et al., 2022; Google, 2023). Recently, those abilities have been extended to multimodal domains, including image, audio, and video (OpenAI, 2024; Gemini Team, 2024a;b). On various professional and academic benchmarks, those models achieve or surpass human-level performance (Hosking et al., 2024). By contrast, the capabilities of pre-trained LLMs in recommendation domain have not derived similar breakthroughs. Traditional recommenders such as matrix factorization (Koren et al., 2009) and sequential item recommenders (Kang & McAuley, 2018) still outperform pre-trained LLMs like Llama (Touvron et al., 2023) by a large margin in domain specific tasks, even after finetuning. One reason is the difference in characteristics of recommendation data and language data. For example, a video recommender system typically recommends videos to users based on their past history of watched or skipped videos, with users rarely providing natural language feedback. We call this *behavioral interaction data*. Recommendation is dependent on this interaction data to train models for recommendation tasks, this data is specifically obtained from the recommender while users are interacting with it. In its native form this data is not textual and most is not available freely on the web, hence off-the-shelf LLMs do not have sufficient understanding of recommendations items & users. Large scale recommendation systems deal with a varied number of items and users, and the interaction data has a very sparse coverage of this large domain-specific vocabulary. Specifically, each user only interacts with a very small set of the full item vocabulary, making it hard to learn representations for the items & users directly using a language model with fine-tuning. For instance, a user will only watch and rate a smaller set of movies from the full catalog of movies.

The sparse nature of this past interaction data makes traditional collaborative filtering (CF) models suitable for tasks in this domain. For example, CF models are good at inferring that if many users have viewed both $v_1$ and $v_2$, then a user who likes $v_1$ may also like $v_2$. Traditional recommenders such as matrix factorization and sequential recommenders outperform very large pre-trained LLMs in such tasks. However, these traditional recommenders do not have good natural language understanding. We expect that combining the language understanding of LLMs and behavioral understanding of

traditional recommenders can help us learn new tasks that utilize language relevant to the domain and vocabulary of domain entities in a unified manner.

Our goal is to improve upon language generation tasks using both behavioral and textual information about an item. We propose Item-Language Model, (**ILM**, hereafter) a framework that learns new input representation that bridge the gap between language domain and recommendation domain to enable new tasks that can utilize both language and behavioral input representations interchangeably in an interleaved manner. Our contribution is adapting Querying Transformer to bridge the gap between language modality and behavioral modality and a new item-item contrastive component in the Querying Transformer to extract behavioral understanding.

## 2 RELATED WORK

**Behavioral representation in LLMs**    Efficiently representing users and items in recommender systems is a rich field with years of work using traditional techniques such as Matrix Factorization (Koren et al., 2009; Rendle et al., 2022). These learn an embedding representation from past interaction data and other metadata about the items. The embedding represents meaningful information extracted about the items & users and projects them in an $N$-dimensional space, with the goal that items & users close together in this space are similar. Let's look at some existing work on representing items & users in LLMs. Using text representation, such as the title of an item or a random identifier to represent recommender users is a straightforward input representation. ELM (Tennenholtz et al., 2024) shows how to interpret input embedding spaces by feeding semantic embeddings and behavioral embeddings to LLM with a Multi-Layer Perceptron (MLP) projection to adapt it to text token space. Similarly, CoLLM (Zhang et al., 2023) feeds user and item collaborative filtering embeddings to LLM to improve quality in recommendation tasks. OpenP5 (Xu et al., 2023; Hua et al., 2023) introduces collaborative indexing techniques that use the structure of the assigned identifier to encode some preprocessed collaborative information. These identifiers are passed, without modification, to the text tokenizer of LLM to improve recommendation tasks. Recently, USER-LLM (Ning et al., 2024) integrates user embeddings within LLMs through a "perceiver" adaptor (Jaegle et al., 2021; Alayrac et al., 2022). This prior work shows that it is hard to improve the pure recommendation capability of LLMs like Llama to match the performance of traditional recommendation-specific models that contain a few transformer layers, trained specifically for recommendation task. Specifically, our work does not tackle the goal of having an LLM beat recommendation task benchmarks. We are interested in enabling new language generation tasks that can use both language and behavioral representations in a unified manner.

**Vision language models**    Work done in computer vision, and specifically, vision-based representations show an alternate approach. Here, vision and language are two different modalities, and the foundation models are trained with both modalities for generative and contrastive learning objectives. Existing work like BLIP-2 (Li et al., 2023), CoCa (Yu et al., 2022), and MaMMUT (Kuo et al., 2023), achieve state-of-the-art performance on vision-language tasks. While these approaches are promising, item representations for recommenders require behavioral data that is usually not public, and thus cannot be used in LLM pretraining. To alleviate this, we adopt a two-phase workflow, similar to the two-phase workflow of BLIP-2, including pretraining a recommender item-adapter in phase-1 and task fine-tuning in phase-2. In addition, we adapt it to include a collaborative item-item contrastive loss.

## 3 PROBLEM SETTING

Please refer to Table 1 for all symbols used in this paper. Consider $H = (H_1, ..., H_N)$ to denote a sequence of inputs to the model. The input data consists of two modalities, text tokens with vocabulary $\mathcal{V}$ or entities (recommendation items $\mathcal{I}$ and users $\mathcal{U}$) with vocabulary $\mathcal{I} \cup \mathcal{U}$. Each item and user is assigned a random identifier and the assigned item ID and user ID are used in the input. $H_i \in \mathcal{V} \cup \mathcal{I} \cup \mathcal{U}$ is the input at position $i$. The order of tokens in the sequential input $H$ contains meaningful information. We want to do well at language tasks by extracting information from the external domain IDs in $H$ and use it by combining with other text inputs. These language tasks are dependent on the IDs. Formally, we want to generate output tokens $O = (O_1, ..., O_M), O \in \mathcal{V}$, such that $O$ performs language tasks using unified understanding of item IDs, user IDs and text inputs in

| Symbol | Description |
|---|---|
| $\mathcal{V}$ | vocabulary of text tokens (from off-the-shelf LLM) |
| $E_v \in \mathbb{R}^{|\mathcal{V}| \times d}$ | embeddings of all text tokens (from off-the-shelf LLM) |
| $d$ | dimension of text token embedding |
| $\mathcal{I}$ | vocabulary of recommendation items (eg, movies) |
| $E_i \in \mathbb{R}^{|\mathcal{I}| \times k}$ | pre-trained behavioral embeddings for all items |
| $\mathcal{U}$ | set of recommendation users |
| $E_u \in \mathbb{R}^{|\mathcal{U}| \times k}$ | pre-trained behavioral embeddings for all users |
| $k$ | dimension of behavioral embedding |
| $H$ | input sequence, $H = (H_1, H_2, \ldots, H_N)$ |
| $O$ | output sequence, $O = (O_1, O_2, \ldots, O_M)$ |

Table 1: Symbols

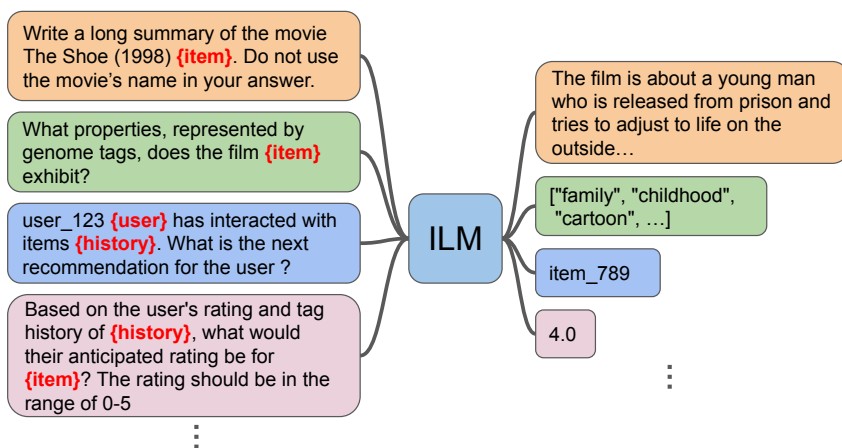

Figure 1: Example tasks in ILM. Recommender domain entities, marked by placeholders in the input, are interleaved with text as input to the model. Where {history} is a sequence of domain items. Some sample outputs are presented respectively for each input

$H$. We assume the pre-trained LLM, a mapping from its text vocabulary $\mathcal{V}$ to text embeddings $E_v$, external domain entities $\mathcal{I}$ and $\mathcal{U}$ are all available to us.

Our technique can be generally applied to any domain by learning embeddings to represent $\mathcal{I}$ and $\mathcal{U}$ from relevant domain data. For the recommendation domain, we learn behavioral embeddings for $\mathcal{I}$ and $\mathcal{U}$ using behavioral information as described in 3.2. These embeddings map $\mathcal{I}$ to $E_i$ and $\mathcal{U}$ to $E_u$. By doing this, we will be able to solve tasks like the ones in Figure 1.

## 3.1 LANGUAGE MODEL

An LLM is trained on large amounts of data, such as billions of words, to learn statistical relationships between words and phrases. This allows them to perform natural language processing tasks, such as generating text, summarizing documents, answering questions, classifying text and learning meaningful representations for text.

In an off-the-shelf LLM the input text is usually broken down into a sequence of language tokens, $l \in \mathcal{V}$, each token is converted into numerical representations called embeddings, $e_l \in \mathbb{R}^d$. The sequence of input embeddings is passed through a stack of decoder layers that are part of a pre-trained LLM and the LLM generates one output token $O_i \in \mathcal{V}$ at a time until a special end-of-sentence token is generated.

In ILM, the input can also be recommendation IDs, $r \in \mathcal{U} \cup \mathcal{I}$. $e_r \in \mathbb{R}^k$ is also available from the recommendation domain. These IDs and text tokens are mixed and appear interchangeably in the input. The spaces $\mathbb{R}^k$ and $\mathbb{R}^d$ are different, and mapping between them is handled in our QFormer

adapter. The QFormer maps ID inputs to language inputs, passed to the LLM. The decoder layers and generation of output tokens $O_i$ are unchanged and reused from off-the-shelf LLM. We introduce technique to generate the embeddings $e_r$ for items and users using behavioral data in next section.

## 3.2 BEHAVIORAL EMBEDDINGS (CF EMBEDDING)

We can swap any embeddings from an external domain. For this paper we utilize collaborative filtering trained using Alternating Least Squares (Rendle, 2022) to generate embedding representations of recommendation items and users. In the recommendation domain, a user $u \in \mathcal{U}$ interacts with item $i$ from a catalog $\mathcal{I}$. We consider all such pairs of $<u, i>$ as a positive interaction examples and all pairs when the user did not interact with the item as negative interaction examples. This data forms a binary matrix of interactions, $A \in \mathbb{R}^{|\mathcal{U}| \times |\mathcal{I}|}$

Formally, *collaborative filtering* (CF, hereafter) does the following, given a matrix of behavioral interactions between users and items, $A \in \mathbb{R}^{|\mathcal{U}| \times |\mathcal{I}|}$, we seek to find matrices $B$ and $C$ such that:

$$A \approx BC, \tag{1}$$

where: $B \in \mathbb{R}^{|\mathcal{U}| \times k}, C \in \mathbb{R}^{k \times |\mathcal{I}|}$

Typically, the scale of $|\mathcal{U}|$ and $|\mathcal{I}|$ varies depending on the domain. The value $k$ is chosen to be much smaller than both $|\mathcal{U}|$ and $|\mathcal{I}|$, resulting in a compressed representation of the original matrix $A$. Hence, these latent representations of the users and items encode rich behavioral information which we will use in our formulation of the ILM to represent recommender items and users.

$e_r \in \mathbb{R}^k$ represents the latent vector for recommender entity $r$ (user ID or item ID).

## 3.3 CO-INTERACTED ITEMS

We define two items $x$ and $y$ as "co-interacted" if the at least one user has interacted with both items, and hence these items have similar representations in the embedding space $E_i$. One interpretation of CF is that the dot-product $e_x.e_y$ represents how similar the two items are. We use this co-interaction signal and the CF embeddings in QFormer as described in the next section.

# 4 ITEM LANGUAGE MODEL

## 4.1 QUERYING TRANSFORMER

We adapt the Querying transformer (*QFormer*, hereafter) of BLIP-2 (Li et al., 2023) as depicted in our Figure 2(a) for the problem of bridging the gap between recommender items modality and text modality. The new component we add to the QFormer is a novel item-item contrastive loss and user-item contrastive loss that preserves the behavioral information in CF embeddings while adapting them to the text modality as depicted in Figure 2(b). The effects of the new component are depicted in Figure 2(c). Our *QFormer* has 4 training tasks. The first 3 tasks are adapted as-is from BLIP-2 and the fourth task is added to extract behavioral information. The possible inputs are pre-trained item CF embedding and/or textual metadata about the item. The actual input and output varies for each task. Our tasks are:

1. Unimodal encoder, which separately encodes an item and text (Unimodal - text is one modality, item ID is another modality). The input is a set of positive item-text pairs, generated from metadata about the item. For example, given a movie and its genre, the movie-genre pairs are "positive" item-text pair. Text from other examples in the batch are used to sample "negative" item-text pairs, for example a movie and a genre that does not belong to it. The text encoder is the same as BERT (Devlin et al., 2019), where a [CLS] token is appended to the beginning of the text input to summarize the text. ***I****tem-****T****ext* ***C****ontrastive* loss (ITC, hereafter) is a contrastive loss that aligns the feature space of the item transformer and the text transformer by encouraging positive item-text pairs to have similar encoded representation in contrast to the negative pairs.

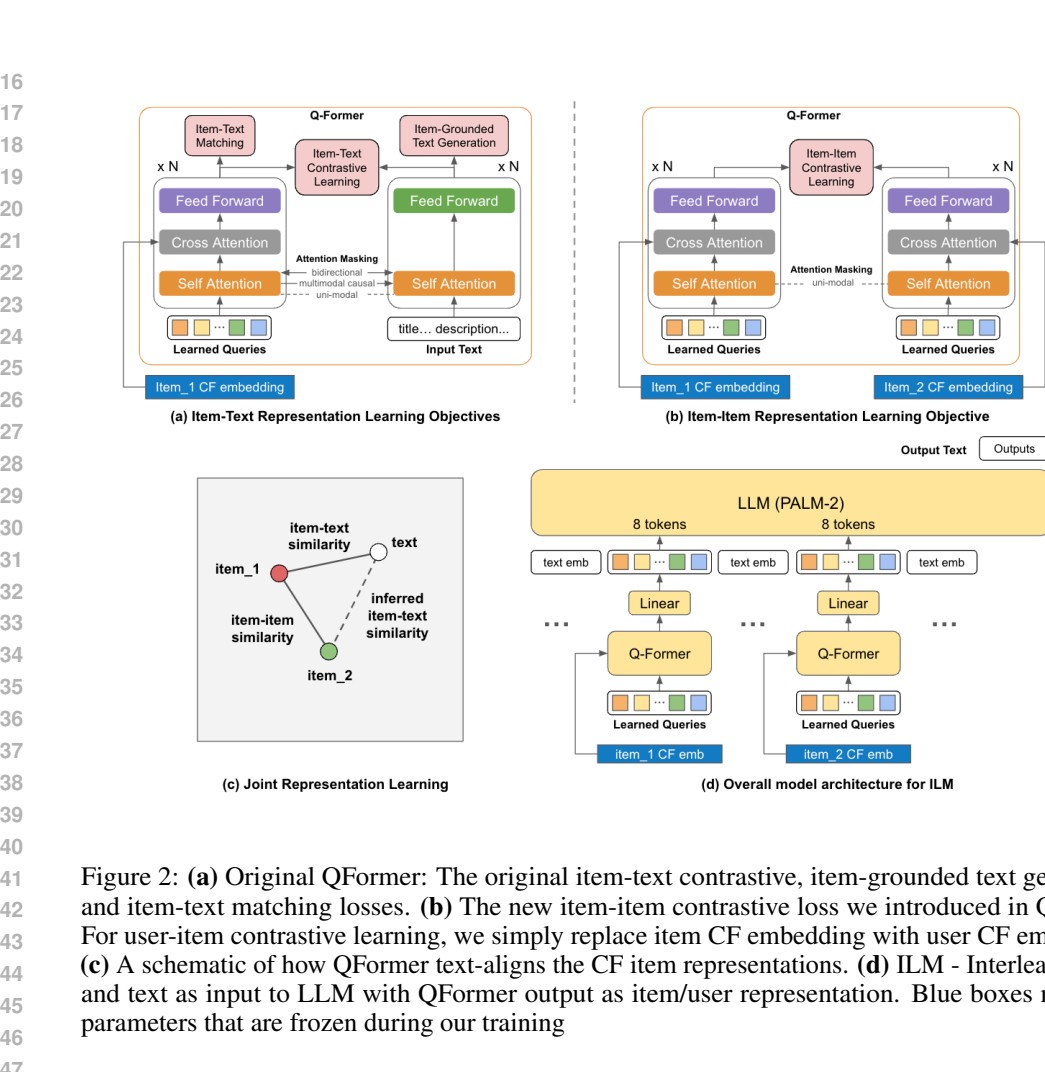

Figure 2: **(a)** Original QFormer: The original item-text contrastive, item-grounded text generation and item-text matching losses. **(b)** The new item-item contrastive loss we introduced in QFormer. For user-item contrastive learning, we simply replace item CF embedding with user CF embedding. **(c)** A schematic of how QFormer text-aligns the CF item representations. **(d)** ILM - Interleaved item and text as input to LLM with QFormer output as item/user representation. Blue boxes mark the parameters that are frozen during our training

2. Item-grounded text encoder, which injects recommender item information by inserting one additional cross-attention layer between the self-attention layer and the Feed Forward Network for each transformer block of the text encoder. A task-specific [Encode] token is appended to the text, and the output embedding of [Encode] is used as the representation of the item-text pair. *Item-Text Matching* loss (ITM, hereafter) aims to learn item-text cross-domain representation that captures the fine-grained alignment between recommendation items and language. ITM is a binary classification task, where the model uses an ITM head (a linear layer) to predict whether an item-text pair is positive (matched) or negative (unmatched) given their input features. For example, this is trained to predict if a movie matches a given genre.

3. Item-grounded text decoder, which replaces the bidirectional self-attention layers in the item-grounded text encoder with causal self-attention layers. A [Decode] token is used to signal the beginning of a sequence, and an end-of-sequence token is used to signal its end. The Language Modeling loss, also called *Item-Text Generative* loss (ITG, hereafter) activates this decoder, which aims to generate textual descriptions given an item. It optimizes a cross entropy loss which trains the model to maximize the likelihood of the text in an auto-regressive manner. For example, given a movie generate the genre tags associated with it or given a movie generate its title.

4. Item-Item unimodal encoder is the new component we add, which separately encodes two co-interacted items and provides one output token per item. *Item-Item Contrastive* (IIC, hereafter) loss aims to preserve behavioral information by encouraging positive item-item co-interacted pairs to have similar representations in contrast to the negative pairs. For example, given a co-interacted pair of movies, their encoder output are trained to be similar

and a pair of unrelated movies are trained to have very different encoder output. This component is reused for *User-Item Contrastive* loss (UIC, hereafter) loss by just replacing the item embedding input with user embedding input. The data for UIC are pairs of <user, item> positive interactions and negative interactions. In our experiments we are focusing on recommender items, but we also show how the framework can be reused for users in the Appendix A.1.

Formally our *Item-Item Constrastive loss* is given by,

$$L = \frac{1}{2N} \sum_{i=1}^{N} \left[ y_i d(x_i, x_i')^2 + (1 - y_i) \max(0, m - d(x_i, x_i'))^2 \right], \qquad (2)$$

where:

- N is the number of items
- $x_i$ and $x_i'$ are QFormer encoder output of two items (co-interacted or unrelated)
- $y_i$ is a binary label indicating co-interacted($y_i = 1$) or dissimilar($y_i = 0$)
- $d(x_i, x_i')$ is the Euclidean distance between $x_i$ and $x_i'$
- $m$ is the margin hyperparameter that defines the minimum distance between dissimilar items.

We can use one or more learned queries per item. *Learned queries* are tokens that are trainable and meant to extract different aspects of information from the item CF embeddings. After this phase, given a CF embedding as input the QFormer will output new representation that is better aligned with language tokens. For each item ID in input the QFormer will output tokens that replace the item ID.

## 4.2 TRAINING

We use pre-trained CF embeddings to represent the domain items and adapt them using QFormer to obtain text aligned input tokens for recommender items. In phase-1 of training, the query tokens in QFormer and the other QFormer layers are trained as part of the four tasks to adapt the frozen CF embeddings to the language domain. We use textual metadata of the items to train these losses, for example the movie title/genre and the movie CF embedding and the co-interacted item pairs to train the QFormer.

Phase-2 trains the full setup including the LLM on language generation tasks as depicted in Figure 2(d). In the item-text mixed input $H = (H_1, ..., H_N)$, item inputs are replaced by the QFormer output tokens. Text inputs are passed directly to pre-trained LLM and tokenized using built-in language tokenizer. Let $\theta$ be the trainable parameters of ILM. Given a downstream loss function $L$ we can differentially optimize the ILM model by solving $\underset{\theta}{\arg\min} L(\text{ILM}(H))$.

## 5 EXPERIMENTS

To assess the method described above, we run a set of experiments on existing baselines to evaluate the generative capability of the Palm-2 LLM with ILM.

**Dataset** We demonstrate the generative capabilities of ILM using all 24 tasks from ELM (Tennenholtz et al., 2024). These tasks are created from the MovieLens 25M dataset (Harper & Konstan, 2015) and consist of 24 movie-focused tasks. The tasks include single movie semantic tasks, such as describing a movie plot or summarizing a movie; single movie subjective tasks, such as writing positive or negative reviews for a movie, and movie pair subjective tasks, such as comparing characteristics of movies. Appendix E in Tennenholtz et al. (2024) provides a complete description of all 24 tasks.

**Setup** We generate two embeddings to represent the items, the *CF embedding* provides a behavioral embedding of the item, this is described in section 3.2, and SentenceT5 (Ni et al., 2022) to obtain a semantic embedding of the item. The title and tags for each movie are used as input to SentenceT5. We then average the resulting output vectors to generate a single semantic embedding for each item. A combined representation of the item using these behavioral and semantic embeddings is

Table 2: Semantic Consistency of ELM (baseline) versus ILM fully finetuned model, using semantic item embedding, behavioral item embedding and combined semantic & behavioral embeddings. Best numbers bolded, next-best underlined

| Tasks | Item Encoder | | | |
|---|---|---|---|---|
| | ELM | ILM-Semantic | ILM-Behavioral | ILM-Combined |
| summary | 81.53 | 82.15 | 74.06 | **82.66** |
| positive review | **88.12** | 87.70 | 79.09 | 87.89 |
| neutral review | 84.41 | 85.10 | 79.44 | **85.48** |
| five pos char. | 86.41 | 90.99 | 82.73 | **91.19** |
| five neg char. | 84.89 | 93.64 | 84.70 | **93.89** |
| long description | 80.81 | 81.15 | 72.58 | **81.58** |
| funnier | 75.52 | 76.10 | 69.43 | **76.78** |
| sadder | 77.86 | 78.66 | 72.04 | **79.39** |
| scarier | 76.77 | 77.96 | 71.99 | **78.50** |
| improve | 83.30 | 84.34 | 79.50 | **84.67** |
| movie to viewer | 84.72 | 88.01 | 79.38 | **88.44** |
| pitch | 87.96 | 88.92 | 83.60 | **89.01** |
| criticize | 83.04 | 84.78 | 80.21 | **85.01** |
| convince1 | 83.02 | **83.66** | 79.20 | 83.60 |
| convince2 | 81.82 | **85.07** | 78.00 | 84.97 |
| convince3 | 80.54 | 84.97 | 77.07 | **85.14** |
| dissuade1 | 80.97 | 81.77 | 78.57 | **81.84** |
| dissuade2 | 80.69 | 85.64 | 79.12 | **85.77** |
| similarities | 84.53 | 90.16 | 80.87 | **90.48** |
| interpolation | 75.94 | 77.85 | 71.92 | **78.38** |
| why like nn | 82.22 | 87.61 | 80.57 | **88.72** |
| diff than nn | 84.70 | 92.57 | 86.51 | **93.28** |
| common with nn | 79.71 | 88.32 | 80.01 | **88.90** |
| all | 82.15 | 85.08 | 78.43 | **85.44** |

paired with textual metadata and co-interacted items to train the phase-1 QFormer tasks ITC, ITG, ITM and IIC with 8 learned query tokens. The phase-2 tasks train the full ILM model along with QFormer model as an adapter for item input, the QFormer generates 8 tokens for each item input. Text inputs are processed by the default PALM 2 text input tokenizer. ILM is trained using the default language model loss and dataset of 24 tasks from ELM. For comparison, the original ELM work used a Multi-Layer Perceptron (MLP) adapter to adapt the item embeddings to language space. In phase-1 they train only the adapter and keep the LLM frozen. In phase-2 they fully train all the parameters in the LLM and adapter.

**Results** We experiment with 3 variants of the setup using semantic embedding of items, behavioral embedding of items and a combination of both semantic and behavioral embedding of items. The results in Table 2 show that semantic embeddings alone perform better than behavioral embeddings, but a combination of both embeddings perform significantly better than semantic embeddings alone. Using behavioral embeddings alone results in poor performance on semantic consistency tasks since behavioral embeddings lack semantic understanding.

In Table 3 we evaluate combined semantic and behavioral embedding model in four different settings,

1. **ILM-MLP** We replace the QFormer with a simple MLP of similar parameter size to evaluate the value of the QFormer architecture, versus a naïve MLP. This is same as the ELM setup, with just one phase of training. Not surprisingly, this performed worse than the original ELM work, as the LLM is frozen.

2. **ILM-Qformer-random** We initialize the QFormer to random values, directly training the output of the QFormer as input of an existing LLM for the final task. We use PALM-2 as the LLM. Note that the LLM is frozen in this setup and there is only one phase of training. This performs better than ILM-MLP, while still worse than the original ELM paper. This

Table 3: Results with various architecture choices. ILM-Qformer-fullyfinetune is costlier but performs the best. ILM-Qformer is a good tradeoff between training cost and performance. Best results bolded

| Tasks | Item Encoder | | | | |
|---|---|---|---|---|---|
| | ELM | ILM-MLP | ILM-QFormer-random | ILM-Qformer | ILM-Qformer-fullyfinetune |
| summary | 81.53 | 78.44 | 80.99 | 81.45 | **82.66** |
| positive review | **88.12** | 85.25 | 85.83 | 85.69 | 87.89 |
| neutral review | 84.41 | 81.47 | 83.96 | 84.24 | **85.48** |
| five pos char. | 86.41 | 85.06 | 85.71 | 85.91 | **91.19** |
| five neg char. | 84.89 | 86.77 | 85.35 | 84.07 | **93.89** |
| long description | 80.81 | 77.83 | 80.05 | 80.19 | **81.58** |
| funnier | 75.52 | 73.42 | 75.23 | 75.93 | **76.78** |
| sadder | 77.86 | 76.06 | 78.12 | 78.18 | **79.39** |
| scarier | 76.77 | 75.24 | 76.99 | 77.08 | **78.50** |
| improve | 83.30 | 77.94 | 82.84 | 83.40 | **84.67** |
| movie to viewer | 84.72 | 80.70 | 84.37 | 84.43 | **88.44** |
| pitch | 87.96 | 85.26 | 88.08 | 88.23 | **89.01** |
| criticize | 83.04 | 79.30 | 83.00 | 82.96 | **85.01** |
| convince1 | 83.02 | 80.74 | 83.46 | 83.03 | **83.60** |
| convince2 | 81.82 | 79.62 | 82.45 | 82.09 | **84.97** |
| convince3 | 80.54 | 77.69 | 81.07 | 80.79 | **85.14** |
| dissuade1 | 80.97 | 79.72 | 81.09 | 80.96 | **81.84** |
| dissuade2 | 80.69 | 80.22 | 81.38 | 80.72 | **85.77** |
| similarities | 84.53 | 83.51 | 85.43 | 85.85 | **90.48** |
| interpolation | 75.94 | 73.86 | 76.95 | 76.78 | **78.38** |
| why like nn | 82.22 | 79.33 | 83.92 | 84.14 | **88.72** |
| diff than nn | 84.70 | 85.09 | 85.48 | 84.54 | **93.28** |
| common with nn | 79.71 | 80.85 | 81.84 | 81.65 | **88.90** |
| all | 82.15 | 80.27 | 82.39 | 82.34 | **85.44** |

demonstrates that while the QFormer has benefit, it alone is not sufficient to beat the existing baseline.

3. **ILM-Qformer** We initialize the QFormer with a phase-1 training. In phase-1, we train the QFormer on ITC, ITG, IIC and ITM losses mentioned earlier. In phase-2, we train the along with a frozen off-the-shelf PALM-2. This performs as well as the ELM model. Note that in the ELM work, all the parameters of the PALM-2 model are fully finetuned. This as a novelty of our paper: the QFormer phase-1 training allows us to skip finetuning the parameters of the LLM, achieving comparable performance at a lower training cost.

4. **ILM-Qformer-fullyfullyfinetune** Similar to ILM-Qformer, but fully finetuning the parameters of the LLM. This performs the best on the evaluation tasks.

These results are consistently observed for semantic embedding model and behavioral embedding model and are attached in the Appendix A.4

To compute **Semantic consistency** (SC), we use the cosine similarity of semantic embeddings of the original target text labels and ILM generated text tokens. Semantic embeddings of the text is obtained by passing the target text labels to Sentence-T5 11B model (Ni et al., 2022). This is based on the original setup described in the ELM paper evaluation setup. The ELM paper does not release its model or evaluation code, hence we reproduce the ELM model and re-report baselines by running the evaluation described in the original paper.

## 6 CONCLUSION

We presented ILM, a novel item-language unified model. We had traditional representations that encode rich information in the recommendation domain and we had language models that provide language understanding, we have shown how we can unify both and learn new tasks that interpolate between these domains and can utilize items and text in a unified fashion. A pre-training step to generate behavioral embeddings is required to ensure our technique performs best. We have shown that we do better when we combine these two domains using semantic consistency tasks from ELM. In Appendix A.1, we used our model designed for language generation task to evaluate hardcore recommendation tasks and show reasonable performance, however the existing baselines for those tasks use different backbone LLMs and are trained to perform well specifically on recommendation tasks and not language generation. We also note that our technique is agnostic to the domain and can be applied to any new domain that has rich embedding representations of domain entities.

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

# A    APPENDIX

## A.1    RECOMMENDATION CAPABILITIES IN LLM

In addition to the main semantic consistency task, we are interested in knowing the domain capabilities added to the LLM as a result of this Item-Language unified architecture. We use a dataset designed for evaluating traditional recommender models. Especially, we are interested in evaluating which aspects of our work contribute to learning domain specific tasks using LLMs.

OpenP5 (Xu et al., 2023; Hua et al., 2023) is a dataset for LLM-based Recommendation development, finetuning, and evaluation. It provides 10 popular preprocessed public datasets, and each dataset contains two kinds of tasks: Sequential Recommendation and Straightforward Recommendation. We select the MovieLens-1M and Beauty datasets for our benchmarks. The training target for each example is the ground truth item ID. For training inputs, we append each item's random indexing ID with its behavioral embedding on the user sequence training set. We use the provided train, development, and test split in the OpenP5 dataset, which uses the last item in the user sequence for testing and the second from the last item in the user interaction sequence for development. For OpenP5 tasks, we report top-k Hit Rate (HR@K) and Normalized Discounted Cumulative Gain (NDCG@K) with $K = 5, 10$ to evaluate the recommendation performance. Since the outputs for the tasks in this dataset are only from the recommender item vocabulary $\mathcal{I}$, to compute those metrics, we use beam search to generate 10 outputs for each example, and remove invalid outputs that do not match the regular expression ".*item_(\d+)$".

## A.2    EFFECTS OF QFORMER PHASE 1 TRAINING.

As shown in Table **??** and Table 7, ILM consistently outperforms ILM-rand by a noticeable margin across all metrics on all benchmarks, which suggests the importance of the QFormer phase-1 training. For the OpenP5 dataset, we experiment with different combinations of phase-1 training losses

1. Only using Item-Text losses (ILM-IT)
2. Combine Item-Text losses with an Item-Item contrastive loss (ILM-IT-II)
3. Combine Item-Text losses with an User-Item contrastive loss (ILM-IT-UI)

We generate item-item pair data for (2) as follows. For each user, we treat two consecutive items in the history sequence as a positive pair, then we perform de-duplication to get all unique pairs as the item-item pair data. The number of pairs generated are shown in Table 5.

The results for the above models are shown in Table 4. We observe that for the Movie Lens 1 Million (ML1M) dataset (Harper & Konstan, 2015), introducing user-item or item-item contrastive losses can, in general, lead to performance gains, while for Beauty there are no obvious gains. We hypothesize this is due to ML1M's item-text pair data being scarce and user interactions are much more richer than in the other two datasets. As can be seen in Table 5, comparing with other datasets, the ML1M dataset contains many fewer users and items, but many more user-item interactions. This supports our hypothesis, and suggests exploring user-interaction signals in the phase 1 representation learning can be beneficial for datasets like ML1M. To demonstrate the regularization effects of the item-item and user-item contrastive losses, we showed the phase 1 final train and eval item-grounded text generation losses in Table 8. We observe that adding item-item or user-item contrastive losses in phase 1 indeed can help to reduce the eval loss and close the train-eval gap.

## A.3    EFFECTS OF NUMBER OF QUERY TOKENS

Another key aspect of our ILM approach is we used multiple learned queries to generate multiple embeddings in QFormer output as item representation to feed into LLM. Existing methods (Tennenholtz et al., 2024; Zhang et al., 2023) typically use one embedding as the item-representation to feed into LLM. We show ILM results using different numbers of queries tokens and a randomly initialized QFormer in Figure 3. In order to better understand the gains of our approach, we also use the MLP approach to project the input embedding into a same number of embeddings. For both approaches, as the number of query tokens increases, the performance first increases then decreases. For most of the query lengths, our method outperforms the MLP approach. Based on this investigation, we chose 8 tokens to present all our results.

Table 4: Effects of phase 1 item-item and user-item contrastive losses on OpenP5 benchmarks

| Methods | ML1M | | | | Beauty | | | |
|---|---|---|---|---|---|---|---|---|
| | HR@5 | NDCG@5 | HR@10 | NDCG@10 | HR@5 | NDCG@5 | HR@10 | NDCG@10 |
| ILM-ITC(seen) | 0.0719 | 0.0474 | 0.1088 | 0.0594 | 0.0212 | 0.0160 | 0.0262 | 0.0177 |
| ILM-ITC-IIC(seen) | 0.0712 | 0.0479 | **0.1093** | **0.0602** | 0.0210 | 0.0160 | 0.0261 | 0.0177 |
| ILM-ITC-UIC(seen) | **0.0724** | **0.0485** | 0.1064 | 0.0595 | **0.0213** | **0.0164** | **0.0270** | **0.0182** |
| ILM-ITC(unseen) | 0.0700 | 0.0470 | 0.1071 | 0.0589 | **0.0218** | **0.0163** | **0.0275** | **0.0182** |
| ILM-ITC-IIC(unseen) | 0.0701 | 0.0472 | 0.1078 | 0.0594 | 0.0216 | 0.0162 | 0.0269 | 0.0180 |
| ILM-ITC-UIC(unseen) | **0.0717** | **0.0481** | **0.1086** | **0.0600** | 0.0213 | 0.0162 | 0.0269 | 0.0181 |

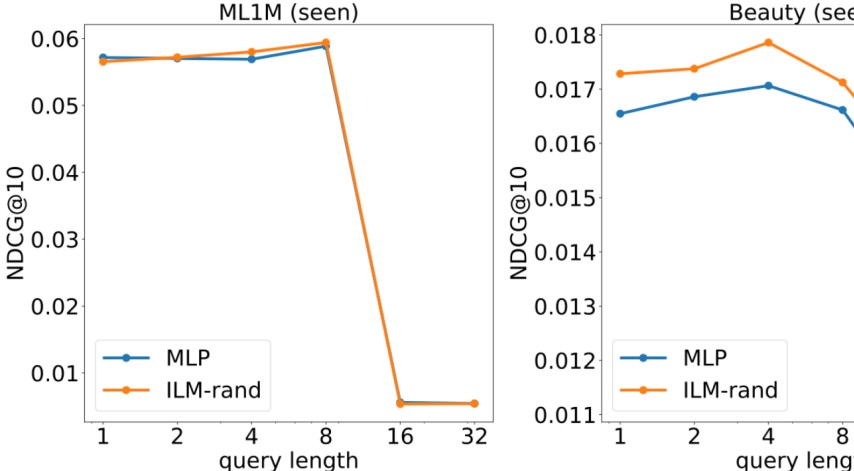

Figure 3: Effects of Number of Query Tokens

Table 5: OpenP5 phase 1 and phase 2 dataset statistics

| Datasets | Phase 1 | | | Phase 2 | | | |
|---|---|---|---|---|---|---|---|
| | Item-text | Item-item | User-item | Train | Test | # Users | # Items |
| ML1M | 3079 | 479664 | 888696 | 19629820 | 12080 | 6040 | 3416 |
| Beauty | 10879 | 103268 | 138521 | 2628260 | 44726 | 22363 | 12101 |

Table 6: Results on OpenP5 sequential recommendation tasks using item behavioral embedding

| Methods | ML1M | | | | Beauty | | | |
|---|---|---|---|---|---|---|---|---|
| | HR@5 | NDCG@5 | HR@10 | NDCG@10 | HR@5 | NDCG@5 | HR@10 | NDCG@10 |
| OpenP5-T5(seen) | 0.2066 | 0.1400 | 0.2945 | 0.1683 | 0.0457 | 0.0336 | 0.0622 | 0.0389 |
| OpenP5-Llama(seen) | 0.0714 | 0.0466 | 0.1094 | 0.0587 | 0.0022 | 0.0036 | 0.0024 | 0.0017 |
| ILM-Qformer-pretrained(seen) | 0.1357 | 0.0910 | 0.1922 | 0.1092 | 0.0227 | 0.0174 | 0.0282 | 0.0192 |
| OpenP5-T5(unseen) | 0.2055 | 0.1386 | 0.2940 | 0.1672 | 0.0452 | 0.0332 | 0.0613 | 0.0384 |
| OpenP5-Llama(unseen) | 0.0556 | 0.0364 | 0.0877 | 0.0467 | 0.0029 | 0.0017 | 0.0045 | 0.0022 |
| ILM-Qformer-pretrained(unseen) | 0.1338 | 0.0902 | 0.1919 | 0.1090 | 0.0220 | 0.0168 | 0.0275 | 0.0186 |

## A.4 SEMANTIC CONSISTENCY RESULTS

We also evaluated ILM model with semantic embedding only and ILM model with behavioral embedding only similar to the results in main section Table 3 on combined embedding model. These results are consistent with the observations on the combined model. ILM-Qformer performs reasonably for a cheaper training cost and ILM-Qformer-fulyfinetune performs the best. We also include the results for semantic only models and behavioral only model in Table 9 and Table 9.

Table 7: Results on OpenP5 straightforward recommendation tasks using item behavioral embedding

| Methods | ML1M | | | | Beauty | | | |
|---|---|---|---|---|---|---|---|---|
| | HR@5 | NDCG@5 | HR@10 | NDCG@10 | HR@5 | NDCG@5 | HR@10 | NDCG@10 |
| HR@5 | | | | | | | | |
| OpenP5-T5(seen) | 0.0347 | 0.0224 | 0.0618 | 0.0309 | 0.0317 | 0.0239 | 0.0437 | 0.0277 |
| OpenP5-Llama(seen) | 0.0106 | 0.0066 | 0.0210 | 0.0104 | 0.0050 | 0.0035 | 0.0065 | 0.0040 |
| ILM-Qformer-pretrained(seen) | 0.0114 | 0.0070 | 0.0241 | 0.0111 | 0.0211 | 0.0161 | 0.0263 | 0.0177 |
| OpenP5-T5(unseen) | 0.0210 | 0.0134 | 0.0303 | 0.0164 | 0.0139 | 0.0089 | 0.0226 | 0.0117 |
| OpenP5-Llama(unseen) | 0.0098 | 0.0066 | 0.0195 | 0.0097 | 0.0047 | 0.0032 | 0.0062 | 0.0038 |
| ILM-Qformer-pretrained(unseen) | 0.0115 | 0.0067 | 0.0250 | 0.0110 | 0.0215 | 0.0162 | 0.0271 | 0.0180 |

Table 8: Effects of phase 1 item-item and user-item contrastive losses on OpenP5 phase 1 final train and eval item-grounded text generation losses

| Methods | ML1M | | Beauty | |
|---|---|---|---|---|
| | Train | Eval | Train | Eval |
| ILM-IT | 0.0000 | 4.1699 | 1.0441 | 4.2643 |
| ILM-IT-II | 0.1552 | 3.8675 | 2.0232 | 3.2567 |
| ILM-IT-UI | 0.0089 | 4.0663 | 2.3420 | 3.3724 |

Table 9: Semantic Consistency (SC) metrics on the ELM 24 tasks using item semantic embedding (PALM2-XS). We define SC as the semantic embedding cosine similarity between the decoded text and original text. We adopt the Sentence-T5 11B model (Ni et al., 2022) for computing semantic embeddings

| Tasks | Item Encoder | | | | |
|---|---|---|---|---|---|
| | ELM | ILM-MLP | ILM-QFormer-random | ILM-Qformer | ILM-Qformer-fullyfinetune |
| summary | 81.53 | 77.42 | 81.35 | 80.98 | **82.15** |
| positive review | **88.12** | 84.67 | 86.12 | 86.14 | 87.70 |
| neutral review | 84.41 | 80.16 | 84.12 | 83.80 | **85.10** |
| five pos char. | 86.41 | 85.02 | 85.58 | 86.17 | **90.99** |
| five neg char. | 84.89 | 86.14 | 84.43 | 84.66 | **93.64** |
| long description | 80.81 | 76.76 | 80.37 | 80.21 | **81.15** |
| funnier | 75.52 | 72.41 | 75.89 | 75.37 | **76.10** |
| sadder | 77.86 | 74.90 | 78.17 | 77.82 | **78.66** |
| scarier | 76.77 | 74.61 | 77.15 | 77.01 | **77.96** |
| improve | 83.30 | 79.46 | 83.08 | 82.97 | **84.34** |
| movie to viewer | 84.72 | 80.05 | 84.19 | 84.40 | **88.01** |
| pitch | 87.96 | 85.35 | 88.24 | 88.17 | **88.92** |
| criticize | 83.04 | 79.41 | 83.10 | 82.86 | **84.78** |
| convince1 | 83.02 | 79.86 | 83.31 | 83.23 | **83.66** |
| convince2 | 81.82 | 79.71 | 82.41 | 82.19 | **85.07** |
| convince3 | 80.54 | 77.57 | 81.20 | 80.60 | **84.97** |
| dissuade1 | 80.97 | 79.36 | 81.33 | 81.08 | **81.77** |
| dissuade2 | 80.69 | 80.17 | 81.25 | 81.03 | **85.64** |
| similarities | 84.53 | 82.67 | 85.86 | 85.66 | **90.16** |
| interpolation | 75.94 | 73.68 | 76.79 | 76.74 | **77.85** |
| why like nn | 82.22 | 76.95 | 84.15 | 83.97 | **87.61** |
| diff than nn | 84.70 | 82.68 | 84.38 | 85.47 | **92.57** |
| common with nn | 79.71 | 79.22 | 82.02 | 82.23 | **88.32** |
| all | 82.15 | 79.60 | 82.44 | 82.37 | **85.08** |

Table 10: Semantic Consistency (SC) metrics on the ELM 24 tasks using item behavioral embedding (PALM2-XS). We define SC as the semantic embedding cosine similarity between the decoded text and original text. We adopt the Sentence-T5 11B model (Ni et al., 2022) for computing semantic embeddings. Best results bolded

| Tasks | Item Encoder | | | | |
|---|---|---|---|---|---|
| | ELM | ILM-MLP | ILM-QFormer-random | ILM-Qformer | ILM-Qformer-fullyfinetune |
| summary | **81.53** | 71.47 | 76.09 | 78.81 | 74.06 |
| positive review | **88.12** | 76.39 | 80.79 | 82.75 | 79.09 |
| neutral review | **84.41** | 73.85 | 79.99 | 82.54 | 79.44 |
| five pos char. | **86.41** | 80.20 | 83.26 | 84.98 | 82.73 |
| five neg char. | **84.89** | 83.43 | 84.46 | 83.70 | 84.70 |
| long description | **80.81** | 70.71 | 75.02 | 77.98 | 72.58 |
| funnier | **75.52** | 68.73 | 71.41 | 73.50 | 69.43 |
| sadder | **77.86** | 70.32 | 73.73 | 75.90 | 72.04 |
| scarier | **76.77** | 70.26 | 73.31 | 75.21 | 71.99 |
| improve | **83.30** | 75.60 | 79.43 | 81.44 | 79.50 |
| movie to viewer | **84.72** | 75.71 | 79.97 | 82.20 | 79.38 |
| pitch | **87.96** | 80.52 | 84.51 | 86.29 | 83.60 |
| criticize | **83.04** | 76.21 | 80.38 | 81.89 | 80.21 |
| convince1 | **83.02** | 75.60 | 80.87 | 82.69 | 79.20 |
| convince2 | **81.82** | 75.31 | 79.94 | 81.77 | 78.00 |
| convince3 | **80.54** | 73.88 | 78.47 | 80.35 | 77.07 |
| dissuade1 | **80.97** | 76.15 | 79.50 | 80.23 | 78.57 |
| dissuade2 | **80.69** | 77.36 | 80.58 | 80.92 | 79.12 |
| similarities | **84.53** | 79.05 | 80.50 | 84.00 | 80.87 |
| interpolation | **75.94** | 71.14 | 71.61 | 74.75 | 71.92 |
| why like nn | **82.22** | 75.76 | 77.52 | 81.06 | 80.57 |
| diff than nn | 84.70 | 80.59 | 81.89 | 84.10 | **86.51** |
| common with nn | **79.71** | 76.51 | 78.76 | 80.57 | 80.01 |
| all | **82.15** | 75.59 | 78.92 | 80.87 | 78.43 |

