# OpenReview forum: "Item Language Model"
_ICLR.cc/2025/Conference — Submitted to ICLR 2025_

### Official Review · Reviewer_45xA · 2024-10-28

**Soundness:** 3
**Presentation:** 2
**Contribution:** 2
**Rating:** 3
**Confidence:** 3

**Summary:**

This work proposes an item language model, which aims to narrow the gap between semantic information in LLMs and behavioral information in embeddings of CF models. Specifically, the authors adopt a Querying Transformer as an adaptor to build the bridge from collaborative information to textual information that could be handled by LLMs. Moreover, they also provide an item-item CL loss for better training. The proposed framework is adopted on PALM-2 and have been evaluated on 24 tasks from ELM, showing superior results compared to ELM.

**Strengths:**

1)	This work adopts Querying Transformer with an additional item-item CL loss to fuse the behavioral embeddings into the language model semantic space, which is sound and effective.
2)	The writing is clear and well organized.
3)	The authors give in-depth model analyses in Appendix.

**Weaknesses:**

1)	The main weakness of this work locates in the experiments. In Table2, comparing ILM-Semantic with ILM-Combined, we only find that there is only marginal improvement in the semantic-behavioral combined strategy. Is this improvement significant (maybe a significance test is needed)? Moreover, it also implies that the behavioral information is not that beneficial for the tasks in ELM (which are mostly language modeling related tasks).
2)	The technical novelty is rather limited, since both the Querying Transformer and the item-item CL are not novel.
3)	The central goal of this work is to discover the best method to fuse behavioral and semantic representations for items in LLMs. However, there are lots of works that have proposed different ways to build item representations in LLM4Rec, which could be discussed in this work.
4)	The authors have shown the results of sequential recommendation in Appendix. We find that the proposed framework cannot outperform OpenP5 in SR tasks. Hence, the possible application scope of ILM is limited (in this work, only the 24 tasks in ELM, which is niche).
5)	Further analyses on the Querying Transformer should be given in the main content. For example, the ablation study of different losses should be given (besides IT, IIC, UIC). Moreover, from Table 4 we can find that the benefits of IIC and UIC are also marginal (significance test is needed), which also implies that these two losses are not that beneficial. Similar experiments are also suggested to be conducted in the main evaluation tasks.
6)	Typos, e.g., Page8, fullyfullyfinetune -> fullyfinetune.

**Questions:**

1) In Figure 3, why does the result drop so dramatically at 8? The optimal query length will be impacted by which factors (e.g., data size)?

---

> ### Author Response · Authors · 2024-11-15
>
> Questions:
> 1. In Figure 3, why does the result drop so dramatically at 8? The optimal query length will be impacted by which factors (e.g., data size)?
>
> This is due to overfitting as the number of parameters to represent the item increases. 8 tokens represent the number of tokens for item input in LLM also increases the trainable parameters.  Yes you are right. Data size affects the number of tokens.
>
> Weaknesses:
>
> 1. The main weakness of this work locates in the experiments. In Table2, comparing ILM-Semantic with ILM-Combined, we only find that there is only marginal improvement in the semantic-behavioral combined strategy. Is this improvement significant (maybe a significance test is needed)? Moreover, it also implies that the behavioral information is not that beneficial for the tasks in ELM (which are mostly language modeling related tasks).
>
> ILM-Semantic is trained with Qformer adaptor and item item contrastive losses, including the other Qformer losses. It shows significant improvement over ELM (which uses a DNN adaptor). We believe that there is benefit to using the Qformer to inject item catalog information in LLMs. Especially in industrial scenario these items are private and information about them are not publicly available on the web for LLMs to natively understand the items.
>
> 2. The technical novelty is rather limited, since both the Querying Transformer and the item-item CL are not novel.
>
> These techniques are adapted to the specific problem of injecting item information in the LLM and enabling the LLM to talk about the items.
>
> 3. The central goal of this work is to discover the best method to fuse behavioral and semantic representations for items in LLMs. However, there are lots of works that have proposed different ways to build item representations in LLM4Rec, which could be discussed in this work.
>
> Our emphasis was not to use language models to improve item recommendation. It was the opposite, “can we use behavioral information and inject this in language models to improve the LLMs ability to interpret and talk about these items”. We see this as something that could complement a recommender, it is not meant to replace a recommender and hence next item prediction was not our goal. We sort of captured this in our introduction section “Our goal is to improve upon language generation tasks using both behavioral and textual information about an item.” and also in our Related work section “ Specifically, our work does not tackle the goal of having an LLM beat recommendation task benchmarks. We are interested in enabling new language generation tasks that can use both language and behavioral representations in a unified manner.”. Note that this is the reason why we chose Demystifying Embedding Spaces using Large Language Models, ICLR 2024 as our baseline. We do not focus on works that try to solve generative item recommendation, and hence we did not consider item tokenizing methods as our baseline. We will take efforts to rewrite to be more detailed about the goal, and make sure it does not get misinterpreted. We apologize for not being more explicit in defining our goal.
>
> 4. The authors have shown the results of sequential recommendation in Appendix. We find that the proposed framework cannot outperform OpenP5 in SR tasks. Hence, the possible application scope of ILM is limited (in this work, only the 24 tasks in ELM, which is niche).
>
> Our goal is to improve the language generation capabilities of the LLM and enable it to speak about these items. We believe this can enable new capabilities in recommendation domain which can truly leverage the language generation capabilities of LLM
>
> 6. Typos, e.g., Page8, fullyfullyfinetune -> fullyfinetune.
>
> Apologies, we will correct this.

---

> > ### Comment · Reviewer_45xA · 2024-11-25
> > **After rebuttal**
> >
> > Thanks for the authors' rebuttals, which have answered some of my questions.
> > My main concern still exists in the weaknesses of novelty and evaluation results/analyses. I will maintain my original voting. I also suggest that the authors could add these discussions to their future version to highlight the motivation of this work.

---

### Official Review · Reviewer_ajDU · 2024-11-02

**Soundness:** 2
**Presentation:** 2
**Contribution:** 1
**Rating:** 3
**Confidence:** 3

**Summary:**

The paper proposes Item Language Model (ILM), a framework to bridge behavioral embeddings from recommendation systems with language understanding in Large Language Models (LLMs). The key innovation is adapting a Querying Transformer (QFormer) with a novel item-item contrastive loss to enable interleaved processing of behavioral and textual information. The framework allows unified handling of recommendation items and text for various language generation tasks.

**Strengths:**

1. The paper addresses a significant gap between behavioral understanding in recommendation systems and language understanding in LLMs. The proposed solution of using QFormer with item-item contrastive learning presents an innovative approach to bridging this modality gap.
2. The addition of item-item contrastive loss to the QFormer architecture is well-justified and demonstrates clear benefits in preserving behavioral information while adapting to the language domain.
3. The ablation studies effectively demonstrate the value of each component, particularly showing how the combination of semantic and behavioral embeddings outperforms using either alone. The comparison across different architectural choices (MLP vs. QFormer vs. QFormer with pre-training) provides empirical support.

**Weaknesses:**

1. The author should further discuss the comparison with existing recommendation systems based on large language models, such as LC-Rec [1] considering compressing product descriptions into a vector input into the large model, BinLLM [2] compressing the user product collaborative signal into an encoded input into the large model, which can also bridge the gap between semantic and recommendation collaborative signals in the large model.

[1] Zheng B, Hou Y, Lu H, et al. Adapting large language models by integrating collaborative semantics for recommendation[C]//2024 IEEE 40th International Conference on Data Engineering (ICDE). IEEE, 2024: 1435-1448.

[2] Zhang Y, Bao K, Yan M, et al. Text-like Encoding of Collaborative Information in Large Language Models for Recommendation[J]. arXiv preprint arXiv:2406.03210, 2024.

2. This is a paper for recommender systems, the author does not seem to have compared the model with a large number of recommendation algorithms, such as classic collaborative filtering, LightGCN, NGCF, or with recommendation algorithms based on large language models, such as LC-Rec, BinLLM, etc. Suggest the author to conduct a more detailed baseline comparison and analysis.

3. The author does not seem to have conducted an analysis of time complexity or the time consumed in inferring user intended products, which reduces the likelihood of using this model in industry.

4. Using the method designed in the article, generate an 8-bit code for each item. Will there be similar products with the same code, which may lead to conflicts in item IDs.

5. The structure and sentences of the paper can be appropriately modified and adjusted to enhance the logic and clarity. Some typos in the paper need to be corrected, such as the missing punctuation mark at the end of the formula sentence in 3.2, some Table reference errors in Appendix.

**Questions:**

Please see the weaknesses.

---

> ### Author Response · Authors · 2024-11-15
>
> We thank you for the detailed review. Please find the answers below
>
> 1. The author should further discuss the comparison with existing recommendation systems based on large language models, such as LC-Rec [1] considering compressing product descriptions into a vector input into the large model, BinLLM [2] compressing the user product collaborative signal into an encoded input into the large model, which can also bridge the gap between semantic and recommendation collaborative signals in the large model.
>
> Our emphasis was not to use language models to improve item recommendation. It was the opposite, “can we use behavioral information and inject this in language models to improve the LLMs ability to interpret and talk about these items”. We see this as something that could complement a recommender, it is not meant to replace a recommender and hence next item prediction was not our goal. We sort of captured this in our introduction section “Our goal is to improve upon language generation tasks using both behavioral and textual information about an item.” and also in our Related work section “ Specifically, our work does not tackle the goal of having an LLM beat recommendation task benchmarks. We are interested in enabling new language generation tasks that can use both language and behavioral representations in a unified manner.”. Note that this is the reason why we chose Demystifying Embedding Spaces using Large Language Models, ICLR 2024 as our baseline. We do not focus on works that try to solve generative item recommendation, and hence we did not consider item tokenizing methods as our baseline. We will take efforts to rewrite to be more detailed about the goal, and make sure it does not get misinterpreted. We apologize for not being more explicit in defining our goal.
>
> 2. This is a paper for recommender systems, the author does not seem to have compared the model with a large number of recommendation algorithms, such as classic collaborative filtering, LightGCN, NGCF, or with recommendation algorithms based on large language models, such as LC-Rec, BinLLM, etc. Suggest the author to conduct a more detailed baseline comparison and analysis.
>
> We will take efforts to rewrite to be more detailed about the goal. We focus more on language generation tasks and improving those tasks by injecting item catalog knowledge in the LLM. Hence we employ Qformer to bridge the gap between item representation and language representation. We do not consider this as a recommender model or a next item predictor. We think this model can complement a recommender by adding language generation capabilities.
>
> 3. The author does not seem to have conducted an analysis of time complexity or the time consumed in inferring user intended products, which reduces the likelihood of using this model in industry.
>
> Sorry for not being more detailed in the definition of our goal. Our goal is language generation as opposed to item prediction. We use collaborative information to improve the language generations tasks as a result of improved item understanding in the LLM. We will take efforts to rewrite and be more clear.  We do see the technique of Qformer and item item contrastive losses performs well when provided with industry scale collaborative information data.
>
> 4. Using the method designed in the article, generate an 8-bit code for each item. Will there be similar products with the same code, which may lead to conflicts in item IDs.
>
> Some of the ELM tasks consider semantic similarities and differences. For example, “Similarities” “interpolation”. These are computed based on the LLM generated output text for each item and hence can be considered as representation of recommender item understanding in LLMs
>
> 5. The structure and sentences of the paper can be appropriately modified and adjusted to enhance the logic and clarity. Some typos in the paper need to be corrected, such as the missing punctuation mark at the end of the formula sentence in 3.2, some Table reference errors in Appendix.
>
> Apologies, we will correct these mistakes and be more elaborate and explicit in defining our goal.

---

> > ### Comment · Reviewer_ajDU · 2024-11-27
> >
> > Thanks for the author's detail response. I will maintain my voting.

---

### Official Review · Reviewer_MDP8 · 2024-11-04

**Soundness:** 3
**Presentation:** 3
**Contribution:** 2
**Rating:** 6
**Confidence:** 4

**Summary:**

The author proposes the Item-Language Model framework, aiming to bridge the gap between item semantic understanding and user behavior learning in recommender systems. By using the Querying Transformer model with generation tasks, the ILM framework can effectively represents the textual and behavior knowledge for better recommendation.

**Strengths:**

- Although it is not the first language-recommendation alignment paper, the authors propose a new direction to integrate these two types of knowledge.
- The ILM-Qformer architecture is superior to a simple MLP model, reflecting the value of QFormer.
- Comprehensive experiments demonstrate the effectiveness of the ILM framework.

**Weaknesses:**

- Is it practical in the industrial scenario where the number of items and users will reach 1 million or even billion? If not, how to solve this problem?
- There are several related works, including:
  - Adapting Large Language Models by Integrating Collaborative Semantics for Recommendation
  - EAGER: Two-Stream Generative Recommender with Behavior-Semantic Collaboration
  - Learnable Tokenizer for LLM-based Generative Recommendation
  - STORE: Streamlining Semantic Tokenization and Generative Recommendation with A Single LLM
What are the difference between ILM and these works? Any experimental comparison?

**Questions:**

Please refer to the weaknesses.

---

> ### Author Response · Authors · 2024-11-15
>
> Thank you for the review, please find the responses to the questions below
>
> 1. Is it practical in the industrial scenario where the number of items and users will reach 1 million or even billion? If not, how to solve this problem?
>
> Yes the item-item contrastive loss and other losses in Qformer are pairwise losses which scale well to large set of items. In fact, these losses benefit from a larger number of item-user interaction data.
>
> 2. There are several related works, including:
> Adapting Large Language Models by Integrating Collaborative Semantics for Recommendation
> EAGER: Two-Stream Generative Recommender with Behavior-Semantic Collaboration
> Learnable Tokenizer for LLM-based Generative Recommendation
> STORE: Streamlining Semantic Tokenization and Generative Recommendation with A Single LLM What are the difference between ILM and these works? Any experimental comparison?
>
> Our emphasis was not to use language models to improve item recommendation. It was the opposite, “can we use behavioral information and inject this in language models to improve the LLMs ability to interpret and talk about these items”. We see this as something that could complement a recommender, it is not meant to replace a recommender and hence next item prediction was not our goal. We sort of captured this in our introduction section “Our goal is to improve upon language generation tasks using both behavioral and textual information about an item.” and also in our Related work section “ Specifically, our work does not tackle the goal of having an LLM beat recommendation task benchmarks. We are interested in enabling new language generation tasks that can use both language and behavioral representations in a unified manner.”. Note that this is the reason why we chose Demystifying Embedding Spaces using Large Language Models, ICLR 2024 as our baseline. We do not focus on works that try to solve generative item recommendation, and hence we did not consider item tokenizing methods as our baseline. We will take efforts to rewrite to be more detailed about the goal, and make sure it does not get misinterpreted. We apologize for not being more explicit in defining our goal.

---

### Official Review · Reviewer_dFu8 · 2024-11-04

**Soundness:** 2
**Presentation:** 2
**Contribution:** 2
**Rating:** 3
**Confidence:** 4

**Summary:**

The paper presents the Item Language Model (ILM), a framework that bridges the gap between behavioral understanding in recommendation systems and language understanding in Large Language Models (LLMs). The ILM framework adapts a Querying Transformer (QFormer) with a new item-item contrastive loss to improve item-text joint understanding. Experiments validate the effectiveness of the proposed method compared with the ELM baseline.

**Strengths:**

1. Writing. The whole paper is written well and easy to follow.

2. Empirical Results. The paper provides strong empirical evidence that combining semantic and behavioral embeddings in the ILM framework leads to improved performance on semantic consistency tasks.

3. General Applicability. The technique is domain-agnostic and can be applied to any domain with rich embedding representations, making it widely applicable.

**Weaknesses:**

1. Limited Novelty. The proposed method is mainly based on BLIP-2, with two additional contrastive losses. However, such kind of contrastive loss is already widely adopted in many self-supervised learning methods for recommendation [1,2]. In a nutshell, the proposed method seems to be a straightforward application of BLIP-2 on recommendation-language pre-training tasks with marginal novelties.

2. Insufficient Experiments. There is only one baseline in the experiments, i.e., the ELM model. There are many recent works in pre-training collaborative-language models [3,4,5,6]. The authors need to compare the performance of their proposed approach with these models, as well as with collaborative models such as SASRec, DIN, FM, or DCN V2. It would be great to evaluate the performance on widely used metrics such as AUC of ROC, Recall or nDCG. Besides, the authors need to justify why QFormer was chosen as the backbone.

3. Insufficient Analysis. There is no in-depth analysis on a) Why does the proposed method work? Does it improve the alignment or uniformity of the representations? [7] b) How does each loss contribute to the performance lift? Could the authors provide an ablation study? c) What's the insight or takeaway of this paper?

[1]. Self-Supervised Learning for Sequential Recommendation with Mutual Information Maximization. CIKM 2020.

[2]. Contrastive Learning for Sequential Recommendation. ICDE 2022.

[3]. Collaborative Cross-modal Fusion with Large Language Model for Recommendation. CIKM 2024.

[4]. LLaRA: Large Language-Recommendation Assistant. SIGIR 2024.

[5]. CoLLM: Integrating Collaborative Embeddings into Large Language Models for Recommendation.

[6]. DisCo- Towards Harmonious Disentanglement and Collaboration between Tabular and Semantic Space for Recommendation. SIGKDD 2024.

[7]. Understanding contrastive representation learning through alignment and uniformity on the hypersphere. ICML 2020.

**Questions:**

1. Why choose QFormer as the backbone architecture? Do the proposed losses generalize well to other backbones? If so, could you provide additional experiments on other backbones? If not, why?

2. Three of the losses are already mentioned in the BLIP-2 paper, the only additional contribution is the user-item and Item-Item contrastive loss, which are also widely adopted in many existing works. Could the authors provide an ablation study as well as an in-depth study of this loss?

3. Could the authors provide the main insight of this work? What should the readers learn from this work?

---

> ### Author Response · Authors · 2024-11-15
>
> Thank you for the review, please find the responses to the questions below
>
> 1. Why choose QFormer as the backbone architecture? Do the proposed losses generalize well to other backbones? If so, could you provide additional experiments on other backbones? If not, why?
>
> Yes, the ELM baseline used a DNN adaptor rather than a Qformer adaptor. The “ELM” baseline we consider is Demystifying Embedding Spaces using Large Language Models, ICLR 2024. The ELM baseline uses a DNN adaptor and PALM2 backbone. In our work we use QFormer adaptor and PALM2 backbone and show significant improvements.
>
> 2. Three of the losses are already mentioned in the BLIP-2 paper, the only additional contribution is the user-item and Item-Item contrastive loss, which are also widely adopted in many existing works. Could the authors provide an ablation study as well as an in-depth study of this loss?
>
> Yes, please find the ablation of the different losses in Appendix Table 4.
>
> 3. Could the authors provide the main insight of this work? What should the readers learn from this work?
>
> Our emphasis was not to use language models to improve item recommendation. It was the opposite, “can we use behavioral information and inject this in language models to improve the LLMs ability to interpret and talk about these items”. We see this as something that could complement a recommender, it is not meant to replace a recommender and hence next item prediction was not our goal. We sort of captured this in our introduction section “Our goal is to improve upon language generation tasks using both behavioral and textual information about an item.” and also in our Related work section “ Specifically, our work does not tackle the goal of having an LLM beat recommendation task benchmarks. We are interested in enabling new language generation tasks that can use both language and behavioral representations in a unified manner.”. We will take efforts to rewrite to be more detailed about the goal, and make sure it does not get misinterpreted. We apologize for not being more explicit in defining our goal.

---

> > ### Comment · Reviewer_dFu8 · 2024-11-25
> >
> > Thanks for the authors' response, and it resolves some of my concerns. Regarding the goal of "enabling new language generation tasks that can use both language and behavioral representations in a unified manner",  there are also a lot of related works in this field to compare, including LC-Rec, TALLRec, LLaRA, and CoLLM.  Therefore, my main concerns regarding the novelty and experiments still exist, so I will leave my rate unchanged. It would be great to present a detailed discussion and/or empirical evaluation between the proposed method and some well-established LLM-based recommenders, as mentioned above, in the future version.

---

### Meta-Review · Area_Chair_jJNp · 2024-12-18

**Metareview:**

- Scientific Claims and Findings:
  - The paper proposes the development of an item language model that enhances the interpretation of items by incorporating behavioral information into language models.

- Strengths:
   - The paper is well-motivated.
   - The use of Q-Former with an additional item-item contrastive loss is sound.

- Weaknesses:
  - The writing lacks clarity, leading to misunderstandings about the main goal of the paper among almost all reviewers.
  - The paper demonstrates limited novelty and provides insufficient evaluation.

- Most Important Reasons for Decision:
   - The decision is primarily based on the identified weaknesses.

**Additional Comments On Reviewer Discussion:**

The rebuttal effectively clarified the main goal of the paper, which many reviewers unfortunately overlooked, indicating a need for clearer writing. However, concerns about limited novelty and insufficient evaluation persist. The paper would benefit from another round of major revisions.

---

### Decision · Program_Chairs · 2025-01-22

Reject